# New Benzimidazoles Targeting Breast Cancer: Synthesis, Pin1 Inhibition, 2D NMR Binding, and Computational Studies

**DOI:** 10.3390/molecules27165245

**Published:** 2022-08-17

**Authors:** Samira Nashaat, Morkos A. Henen, Shahenda M. El-Messery, Hassan Eisa

**Affiliations:** 1Department of Pharmaceutical Organic Chemistry, Faculty of Pharmacy, Mansoura University, Mansoura P.O. Box 35516, Egypt; 2Department of Biochemistry and Molecular Genetics, University of Colorado Denver, Denver, CO 80204, USA

**Keywords:** benzimidazole, breast cancer, NMR, ^15^N–^1^H HSQC, Pin1 inhibitors, computational studies

## Abstract

Benzimidazole derivatives are known to be key players in the development of novel anticancer agents. Herein, we aimed to synthesize novel derivatives to target breast cancer. A new series of benzimidazole derivatives conjugated with either six- and five-membered heterocyclic ring or pyrazanobenzimidazoles and pyridobenzimidazole linkers were synthesized yielding compounds **5**–**8** and **10**–**14**, respectively. Structure elucidation of the newly synthesized compounds was achieved through microanalytical analyses and different spectroscopic techniques (^1^H, ^13^C-APT and ^1^H–^1^H COSY and IR) in addition to mass spectrometry. A biological study for the newly synthesized compounds was performed against breast cancer cell lines (MCF-7), and the most active compounds were further subjected to normal Human lung fibroblast (WI38) which indicates their safety. It was found that most of them exhibit high cytotoxic activity against breast cancer (MCF-7) and low cytotoxic activity against normal (WI38) cell lines. Compounds **5**, **8**, and **12**, which possess the highest anti-breast cancer activity against the MCF-7 cell line, were selected for Pin1 inhibition assay using tannic acid as a reference drug control. Compound **8** was examined for its effect on cell cycle progression and its ability to apoptosis induction. Mechanistic evaluation of apoptosis induction was demonstrated by triggering intrinsic apoptotic pathways via inducing ROS accumulation, increasing Bax, decreasing Bcl-2, and activation of caspases 6, 7, and 9. Binding to 15N-labeled Pin1 enzyme was performed using state-of-the-art ^15^N–^1^H HSQC NMR experiments to describe targeting breast cancer on a molecular level. In conclusion, the NMR results demonstrated chemical shift perturbation (peak shifting or peak disappearance) upon adding compound 12 indicating potential binding. Molecular docking using ‘Molecular Operating Environment’ software was extremely useful to elucidate the binding mode of active derivatives via hydrogen bonding.

## 1. Introduction

Cancer is considered the most troublesome disease worldwide. It is responsible for almost seven million deaths every year, which is expected to double by 2030 [1,2]. One of the most common cancer types is breast cancer, which is a malignant tumor formed in breast tissues including: lobules, ducts, and the cells surrounding them [3,4]. It is the second most common cancer after lung cancer (15.4% of all cancers) [5], and it is the most common cancer in women [6,7]. For that reason, there is an urgent need for developing new and more effective treatments to reduce its health problem.

Benzimidazole is one of the most important heterocyclic compounds due to its numerous biological activities such as anti-inflammatory [8], antiviral [9,10], antibacterial [11,12], antifungal [13,14] and antioxidant effect [15,16]. Benzimidazole ring is especially relevant for its known anticancer activity against breast cancer [17,18,19,20]. Appendix A is showing important benzimidazole derivatives **A**–**F** with known anti-proliferative activity.

On the other hand, chalcone moiety has received great attention due to its versatile therapeutic activities (especially anticancer activity), and it can be cyclized to give heterocyclic rings like pyrazole and oxazole, or six-membered heterocyclic rings like pyran and pyridine, which in turn possess different biological activities especially concerning anti-breast cancer activity [21,22,23,24]. Additionally, pyrazinobenzimidazoles and pyridobenzimidazole were reported due to the effectiveness of them as anti-breast cancer agents [25,26,27].

Human Pin1 is a member of the family of peptidyl-prolyl *cis-trans* isomerase which binds to proteins containing phosphorylated S/T-P sites. The isomerization of the pSer/pThr-Pro peptide bond can control the function of these proteins in different pathways such as gene promoters, instructs transcription complexes towards specific gene expression profiles, immune response, germ cell development, regulation of cell growth, genotoxic and other stress responses [28,29]. Pin1 can bind to estrogen receptor (ER), and it has a regulatory effect on the disordered AF1 of estrogen receptors alpha [30]; also, it increases the DNA binding to (ER) so that the upregulation of Pin1 leads to different cancers, particularly breast cancer. Therefore, Pin1 is considered an important target for designing novel anti-breast cancer drugs.

Indole-2-carboxylic acid (I) was used as a lead compound (Pin1inhibitor, IC_50_ 16 µM) to design benzimidazole-2-carboxylic acid derivatives (II) (IC_50_ 10–740 µM) of Pin1′s PPIase activity inhibition [31], where additional *N* atom has increased the inhibitory potency of the designed products by forming *H*-bond between the two nitrogen (*N*, *NH*) and Cys113 thiol and the Ser154 alcohol [32] ([Fig molecules-27-05245-ch001]).

In continuation of our previous work targeting breast cancer via Pin1 inhibition [33], herein we design new benzimidazoles by altering the carboxylic group of compound II with different substitutions at the 2-position to give new products with expected improved activity. 

We adopt the molecular hybridization technique to design new hybrids structurally containing benzimidazole core conjugated with different pharmacophores aiming to develop selective and more-effective anti-breast cancer agents via Pin1 inhibition ([Fig molecules-27-05245-ch001]). In a remarkable addition to our study, we further apply ^15^N-^1^H HSQC NMR to detect binding between Pin1 and the newly synthesized compounds which are considered an extremely practical and sensitive tool in drug design.

All the newly synthesized hybrids were evaluated for their in vitro anti-breast cancer activity against breast carcinoma (MCF-7). In vitro cytotoxicity against human normal cell lines was applied. The most active hybrids were evaluated for Pin1 enzyme inhibition. We used several mechanistic studies to explore the mechanism of action: firstly, cell cycle distribution and induction of apoptosis; secondly, different apoptotic mechanisms were evaluated such as gene expression analysis of some apoptotic, anti-apoptotic factors, the expression level of BAX, BCL-2, Cyto c, caspases 6, 7 and 9 in MCF-7 cells. Moreover, reactive oxygen species (ROS) intracellular accumulation mechanisms were all inspected. Computational studies such as docking, surface mapping, and contact preference were studied. 

## 2. Result and Discussion

### 2.1. Chemistry

The target compounds of our work were prepared via two schemes. For Figure 1, the reaction of dichlorophenylenediamine **1** with lactic acid was performed according to the Phillips method, as reported in the literature to yield intermediate compound **2** [26]. Compound **2** was oxidized using K_2_Cr_2_O_7_ to form compound **3** [26]. The newly synthesized chalcone **4** was obtained from compound **3** by Claisen–Schmidt condensation with dimethylaminobenzaldehyde [34]. The structure of the newly synthesized compound has been confirmed by different spectroscopic techniques including IR, MS, ^1^H-NMR, and ^13^C-NMR. The IR spectrum showed strong absorption bands at (1641 cm^−1^) representing the carbonyl group and the ^1^H NMR spectrum showed two doublet peaks at 7.77 and 7.93 ppm with *J* coupling 12.6 Hz representing the two protons of the unsaturated ketone.

One of the most common methods for synthesis of pyrazoline is the reaction of α,β-unsaturated carbonyl compound with hydrazine hydrate [35,36]. Compound **5** was synthesized by refluxing chalcone with hydrazine hydrate in ethanol via aza-Michael addition reaction [37] which was followed by cyclization and dehydration [35]. The ^1^H NMR spectrum of the formed product showed a singlet peak for CH_2_ at 2.9 ppm and the disappearance of the peaks from CH=CH part at 7.77 and 7.93 ppm which confirmed the formation of the pyrazole ring. On the other hand, the ^13^C NMR spectrum showed the disappearance of the C=O peak at 180 ppm and the appearance of 3 carbon of pyrazole ring (one saturated and 2 C=N) at 30.1 and 155.5 ppm, respectively.

Compounds **6** and **7** were synthesized by the reaction of α, and β-unsaturated ketone with malononitrile in the presence of ammonium acetate and piperidine, respectively [36]. Both of the compounds were confirmed by IR, ^1^H NMR, and microanalytical data. For compound **6**, the IR spectrum showed a band at (3225–3356) cm^−1^ characteristic for the (NH_2_) group and a band at 2214 cm^−1^ for the (CN) group and the absence of the (C=O) band, while for compound **7**, IR spectrum showed a band at 3424 cm^−1^ characteristics for the (NH_2_) group and a band at 2208 cm^−1^ for the (CN) group and also the absence of the (C=O) band. The ^13^C NMR spectrum showed an increase in the number of carbon atoms by three carbons than started compound representing the formation of pyran ring and attached nitrile group (CN) and disappearance of the peak of carbonyl at 180 ppm of the started chalcone. In addition, the Attached Proton Test (APT) was performed for compound **7**.

Compound **8** was synthesized by the reaction of chalcone with hydroxyl amine and KOH in absolute ethanol. Its structure was confirmed by ^1^H NMR spectra which showed a doublet peak for CH_2_ of isoxazoline ring at 3 ppm and a triplet peak for CH of isoxazoline ring at 5.75 ppm, while ^13^C NMR spectrum showed the disappearance of carbonyl carbon C=O of the started chalcone appeared at 180 ppm and appearance of two new carbon atoms -C-O and -C-C=N at 62.92 and 46.36 ppm, respectively. Moreover, additional COSY study was used to confirm structure of compound **8.**

Figure 2 focused on the synthesis of the new pyrazinobenzimidazoles (**10**–**13**) and pyridobenzimidazole **14**. Compound **9** was synthesized by the reaction of compound **3** with *p*-bromophenacyl bromide and K_2_CO_3_ in acetone. The formed compound was confirmed by IR, ^1^H NMR and microanalytical data. The ^1^H NMR spectra of the product showed singlet peak of methylene protons at 6.19 ppm, also the IR spectrum showed 1654–1689 cm-^1^ bands belonging to the two (C=O) and the mass spectrometry showed [M + 2] in 1:1 molar ratio indicating the presence of Br atom. Compounds **10**, **11** and **13** were prepared by reaction of compound **9** with p-toluidine, *o*-toluidine and ammonium acetate, respectively, in glacial acetic acid. The prepared compounds were confirmed by the IR spectrum which showed the disappearance of the two carbonyl bands (1654 and 1689) indicate the intramolecular cyclization. Moreover, the mass spectrometry of the compounds showed M^+2^ in 1:1 molar ratio indicating the presence of Br atom.

Compound **12** was synthesized by the reaction of compound **9** with benzylamine in acetic acid and its structure was confirmed by the IR spectrum showing the disappearance of the two carbonyl bands indicating the intramolecular cyclization, also the ^1^H NMR spectrum of the formed product showed a singlet peak corresponding to CH_2_-Ph at 4.64 ppm. Additionally, APT was performed for compound **12**. Finally, compound **14** was synthesized by the reaction of compound **9** with K_2_CO_3_ in butanol. The IR spectrum showed that the disappearance of the two carbonyl bands indicates the intramolecular cyclization, while ^1^H NMR spectra of the formed product showed a singlet peak corresponding to OH at 9.07 ppm. It was confirmed that the structure of the product is adopting a fully unsaturated form of A, not B.

For compounds (**10**–**14**), the ^1^H NMR spectra indicate that the methylene protons of compound **9** resonated in the aliphatic area, and after cyclization, the corresponding proton in compounds (**10**–**14**) shifted to the aromatic region. This deshielding is due to the aromatic ring current (the magnetic field induced by π electrons which reinforces the applied magnetic field, so a higher frequency is needed for resonance, which results in the deshielding of the protons). The physicochemical properties of the newly synthesized compounds (**4**–**14**) are shown in Table 1. 

### 2.2. In Vitro Antitumor Activity

All our newly synthesized hybrids were evaluated for their in vitro anticancer activity by the standard MTT assay [38] against a breast cancer cell line (MCF-7). MTT is a calorimetric assay which relies on the ability of live tumor cells to reduce yellow MTT (3-(4,5-dimethylthiazol-2-yl)-2,5-diphenyltetrazolium bromide) to purple formazan by mitochondrial dehydrogenases, then added DMSO to dissolve the purple formazan and quantified by measuring absorbance at a certain wavelength. The dose response curve for compounds-treated vs. untreated cells is used to quantify the amount of formazan produced to report on the effectiveness of the compounds. The obtained results revealed that the tested compound **8** exhibited a very strong potency against the tested breast cancer cell line with IC_50_ at 8.76 µg/mL compared to Doxorubucin (reference IC_50_ at 4.17 µg/mL). Additionally, compounds **5** and **6** have shown strong activity with IC_50_ at 11.78 and 15.48 µg/mL, respectively. Moreover, Compounds **4**, **7**, **11**, **12**, and **14** showed moderate activity against the same cell line with IC_50_ ranging from 21–50 µg/mL. On the other hand, compounds **9**, **10**, and **13** showed the lowest potency in **13** exhibited the least inhibitory activity with IC_50_ at 71.88 µg/mL. The cytotoxicity activity of some compounds against MCF-7 is summarized in Table 2. 

### 2.3. In Vitro Cytotoxicity against Human Normal Cell Line

To investigate whether the newly synthesized compounds exhibited selective activity against cancer and normal cells, compounds **5**, **8**, and **12** which displayed high anti-breast cancer (MCF-7) activity were selected for screening for their cytotoxic activity on normal Human lung fibroblast (WI38) cell line to determine their therapeutic safety. As shown in (Table 3), all tested compounds displayed low cytotoxicity against normal (WI38) cell line, in which compound **8** exhibited the lowest cytotoxicity against (WI38) with IC_50_ 71.62 µM in comparison to Dox (6.72 µM).

### 2.4. Pin1 Inhibition Assay

As shown in (Table 4), compounds **5**, **8**, and **12** which exhibited the highest anti-breast cancer activity against the MCF-7 cell line were selected for Pin1 inhibition assay against tannic acid as reference drug control. The results demonstrated that all the compounds tested exhibited a very strong pin1 inhibitory activity in which the most active one is compound **5**. 

Addition compound **12** which shows good Pin1 inhibition activity with IC_50_ 1.106 nM, was subjected to ^15^N–^1^H HSQC NMR analysis bound to Pin1 enzyme in which the overlay showed CSP and peak disappearance for different peaks as indicative of binding on different time scales.

### 2.5. NMR Binding Technique (2D ^15^N-^1^H HSQC)

2D ^15^N-^1^H HSQC NMR is a very sensitive technique for the detection of binding between specific biomolecule and a small ligand. Not only can it report binding, HSQC spectra can also report the time regime of exchange between free and bound form if there’s binding. 

Herein, ^15^N-labeled Pin1 enzyme is used where we recorded HSQC spectra of the protein alone and in the presence of an excess amount of one of the newly synthesized compounds. The main challenge we faced in our research was the compound solubility in a balanced mixture of the protein aqueous buffer. Although our compound was freely soluble in DMSO, it is inapplicable to use pure DMSO to dissolve the protein because this will result in protein denaturation and probably aggregation. Therefore, only one compound was tested where we were able to dissolve it partially in a mixture of 95% buffer and 5% DMSO. In addition, as a control, the spectrum was recorded only in presence of DMSO and compared to the ones with the ligand. The proportion of Pin1 to the compounds was adjusted to be 1:10. The recorded spectra demonstrated chemical shift perturbation for some of the amino acids in the protein (CSP). Chemical shift perturbation and peak shifting are indicative of fast exchange between free and bound states which results in peak positions averaging. Moreover, some peaks corresponding to different amino acids showed peak disappearance upon binding. This peak disappearing indicated exchange on the μs–ms time scale (Figure 1). Furthermore, our NMR results have been verified using cell lines.

### 2.6. Apoptosis Analysis of Compounds **8**

According to the results from the cell-based assay and NMR binding to Pin1, we chose compound **8** to additional estimate its effect on the apoptotic process. MCF-7 cells were treated with compound **8** at a concentration of 2 µM for 24 h and processed for flow cytometry in the presence of Annexin V-FITC/PI and were investigated. As shown in (Table 5), cells treated with compound **8** showed higher DNA content at the Pre-G1 phase (38.25% vs. 1.47% for the control) corresponding to the sum of the percentage of Annexin V-positive/PI-negative staining and double-positive staining cells (early and late apoptosis, respectively). Figure 2 shows the flow cytometry results.

### 2.7. Annexin V-Fluorescein Isothiocyanate (FITC)/PI Dual Staining Assay

This assay was performed using a breast MCF-7 cell line to determine the percentage of apoptosis induced by compound **8***,* in which MCF-cells were incubated with compound **8** at 2 µM concentration for 24 h [39]. Compound **8** induced early apoptosis (3.7%) in MCF-7 and improved apoptosis (15.32%) by 128-fold over untreated cells, and induces apoptosis with values of (38.25%) (Table 6 and Figure 3 and Figure 4); these values were inconsistent with data obtained from DNA content.

### 2.8. Measuring the Expression of Apoptotic and Anti-Apoptotic Markers

Apoptosis can be excited by different stimuli via intrinsic and extrinsic pathways [40]. It was reported that pin 1 inhibition involving apoptosis was performed via intrinsic mitochondrial l) pathways [41,42]. The mitochondrial apoptotic pathway was mediated by several gene products including the cl-2 family member which is comprised of pro-apoptotic (e.g., Bax, Bad) and anti-apoptotic (e.g., Bcl-2, Bcl-w) proteins that undergoes a conformational change in mitochondrial membranes, hence, mediates the release of cytochrome c from the intermembrane space into the cytosol [43]. The Bax/Bcl-2 ratio plays an important role in controlling the release of mitochondrial cytochrome c [44], after its release to the cytosol, caspase 9 is activated which can mediate downstream caspase activation including: caspase-3, caspase-6, and caspase-7 that obligated cells to apoptosis [45]. Based on the above, we examined compound **8** capability to trigger intrinsic apoptotic pathways by affecting the expression of these markers in MCF-7 cells. Demonstrated in (Figure 5), compound **8** increased the level of proapoptotic proteins: Bax by 4.63-fold and decreased the levels of anti-apoptotic protein Bcl-2 by 39% compared to the control. In addition, it elevated the level of cytochrome C by about 2.49-fold compared to the control. Furthermore, compound **8** enhanced the level of caspase 6, 7 and 9 by about 1.16, 9.06, and 8.97-fold, respectively, compared to the control cell. Therefore, from previous data we concluded the compound induces apoptosis in the breast MCF-7 cell through the mitochondrial-mediated pathway. The effect values of compound **8** on the expression level of BAX, Bcl-2, Cyto c, Casp 6, 7 and 9 in MCF-7 cells are listed in Table 7.

### 2.9. Intracellular ROS Accumulation Assay

Reactive oxygen species (ROS) are chemically reactive species containing oxygen atoms including peroxides, superoxide, and hydroxyl radicals. They play an important role in the activation of apoptosis under physiological or pathological conditions. Direct or indirect ROS action can mediate the release of mitochondrial cytochrome c which trigger the induction of caspase [46]. The intracellular level of ROS was estimated by The Human ROS ELISA Kit, which is based on standard sandwich enzyme-linked immunosorbent assay technology. It was found that compound **8** induces ROS accumulation in MCF-7 cells by 1.258-fold higher than the control cells (Figure 6); these results indicated that compound **8** induces the apoptosis of the MCF-7 cell line by ROS-mediated mitochondrial pathway.

### 2.10. Computational Study

#### 2.10.1. Conformational Analysis

Conformational analyses of compounds, **5**, **8**, and **12** have been performed. The best energy conformers were obtained by conformational searching using the multi-conformer method (Figure 7). Further in-depth docking study was performed for the aforementioned compounds which showed good binding to Pin1 enzyme via NMR biding technique.

#### 2.10.2. Molecular Docking Study

We chose the benzimidazole derivative co-crystallized in Pin1 crystal structure obtained from the protein data bank (PDB: 4TYO) as a reference for our docking study because of the similarity of its benzimidazole ligand core structure to our compounds [31]. This reference ligand showed a tight binding via the interaction with a network of hydrogen binding with Arg 69, Lys63, and Cys113 in addition to cationic hydrogen binding with both Phe134 and Ser115 residues (Figure 8). It was reported that the interaction with Lys63 is essential for Pin1 inhibition [31].

Compound **8** is well-positioned to interact with Lys63 and Arg69 amino acid residues via hydrogen bonding interaction with *N* and *O* atoms, while compound **12** is well-positioned to interact with Lys63 amino acid residue via the phenyl ring (Figure 9).

Compound **5** has shown both hydrogen bonding and n-cationic interactions to Arg69 and Phe134, respectively (Figure 10a). The aligned conformation of compound **5** in the binding site was aligned completely inside the binding site surface map explaining its activity experimentally determined by NMR binding technique (Figure 10b).

#### 2.10.3. Contact Preference

The purpose of the Contact Statistics application is to calculate, from the 3D atomic coordinates of ligand, preferred locations for hydrophobic and hydrophilic ligand atoms. In particular, this work aimed to study the interactions between the chemical components of the ligands and the protein microenvironment surrounding them. The results obtained indicate that the information underlying the fragment contacts is valuable as it demonstrates the similar pattern of distribution of the hydrophobic and hydrophilic sites between our ligands **5**, **8**, and **12** as shown in (Figure 11) and can also be exploited in understanding results of molecular docking simulations. A better understanding of the interaction patterns of these moieties can lead to the improved application of ligand binding prediction, protein function recognition, and drug design tools.

#### 2.10.4. Surface Mapping

To confirm the similarity between our Pin1 active candidate binders, we performed enzyme surface mapping. (Figure 12) showed nearly typical surface mapping contours of selected ligands. This similarity in surface mapping could be contributing to their good binding to Pin1 and put more evidence that their anti-breast cancer activity may be attributed to Pin1 inhibition.

### 2.11. Structure-Activity Relationship

For Figure 1: All the compounds showed very strong to moderate cytotoxic activity in which compound **8** is the most active with IC_50_ of 8.76 μM against MCF-7 and compound **4** is the least active with IC_50_ of 26.87 μM against MCF-7. Moreover, we observed that compounds with five-membered heterocyclic rings **6** and **7** are more potent than compounds with six heterocyclic rings **5** and **8**. So the cyclization strategy is considered a successful one in increasing the anti-breast cancer activity. 

For Figure 2: Pyrazanobenzimidazole compounds showed moderate **11** and **12** to weak cytotoxic activity **10** and **13** IC_50_ values ranging from 31.30 μM to 71.88 μM against MCF-7.

## 3. Conclusions

The design, synthesis, and biological evaluation of new hybrids targeting Pin1 inhibition as anti-breast cancer agents were performed. The structure of the prepared compounds was confirmed by IR, ^1^H NMR, ^13^C NMR, APT (for compounds, **7**, **12**), and COSY (for compound **8**). Anti-proliferative activity of newly synthesized compounds was evaluated against breast cancer MCF-7 cells lines and the results demonstrated that compounds **5**, **8** and **12** showed the highest activity with IC_50_ 11.78, 8.76 and 31.30, respectively, so these compounds were screened for their cytotoxicity against normal Human lung fibroblast (WI38) that gives an indication of their therapeutic safety, and they exhibit low cytotoxic activity denoted from their IC_50_. Structure-activity relationship have shown that cyclization of chalcone derivative is a successful strategy in designing of anti-breast cancer agent. Compounds **5**, **8**, and **12** also were selected for Pin1 inhibition assay against tannic acid as reference drug control. Remarkable multidimensional NMR state-of-the-art binding studies has been performed where ^15^N–^1^H HSQC NMR spectra were done for Pin1 bound to compound **12** in which the spectra overlay showed chemical shift perturbation and peak disappearance for different peaks indicative of binding on different time scales. Cell cycle analysis of MCF-7 cells treated with compound **8** showed higher DNA content at the Pre-G1 phase. The pro-apoptotic activity of **8** was indicated by the significant increase in the percentage of annexin V-FITC-positive apoptotic cells. Mechanistic evaluation of apoptosis induction was demonstrated by triggering intrinsic apoptotic pathway via inducing ROS accumulation, increasing Bax, decreasing Bcl-2 and activation of caspases 6, 7 and 9. Molecular modeling study of the most active compounds **5**, **8** and **12** against Pin1 inhibition demonstrated good agreement with the obtained biological result. Different molecular modeling studies of the most active compounds **5**, **8** and **12**, including docking against Pin1 demonstrated good agreement with the obtained biological result. In addition to contact preference and surface mapping studies have been performed.

## 4. Experimental Work

The synthesis of the designed compounds was performed at the Faculty of Pharmacy, Mansoura University, Mansoura, Egypt. All of the new compounds were analyzed for C, H and N and agreed with the proposed structures within ±0.4% of the theoretical values. Melting points (°C) were determined on Mettler FP80 melting point apparatus and are uncorrected. IR spectra were determined for KBr discs on Thermo Fischer Scientific Nicolet IS10 Spectrometer (υ in cm^−1^) at the Faculty of Pharmacy, Mansoura University, Egypt. The ^1^ H NMR spectra were obtained in DMSO-*d_6_* or CDCl_3_ by using Bruker 400 MHz or Jeol 500 MHz at the Faculty of Science, Mansoura University, Egypt using TMS as internal standard (chemical shifts in ppm). The ^13^C NMR spectra were obtained in DMSO-d_6_ by using Jeol 500 or Bruker 400 MHz at the Faculty of Pharmacy, Mansoura university, Egypt using TMS as the internal standard (chemical shifts in ppm). The completion of reactions was monitored using TLC plates, Silica gel 60 F_254_ precoated (E.Merck) and the spots were visualized by UV (366 nm) and KMnO_4_. CH_2_Cl_2_:MeOH (10:1) and pet. ether:EtOAc (1:1) or (3:1) were adopted as elution solvents. The in vitro anticancer screening was conducted in the Faculty of Pharmacy, Mansoura University, Mansoura, Egypt. Mass spectrometry (MS) data were obtained on a Perkin Elmer, Clarus 600 GC/MS, and Joel JMS-AX 500 mass spectrometry. Molecular docking experiments were performed using ‘Molecular Operating Environment’ software on a Core i7 workstation.

### 4.1. Chemistry

#### 4.1.1. Preparation of (*E*)-1-(5,6-Dichloro-1*H*-benzo[d]imidazol-2-yl)-3-(4-(dimethylamino)phenyl)prop-2-en-1-one **4**

A mixture of ketone **3** (0.2 g, 1 mmol) and dimethylaminobenzaldehyde (1.5 mmol) in ethanol (95%, 15 mL) was reacted at room temperature for 15 min in an ice bath. A solution of NaOH (3.0 g) in ethanol (95%, 10 mL) was added dropwise with continuous stirring for 10 min, the reaction mixture was continuous stirring for 24–72 h, the resulted solution was poured in ice water, neutralized by dilute HCl, filtered the formed solids, washed with water and dried to give the desired compound as dark brown to black solids with mp = 258–260 °C (yield 95%).

Spectral data: ^1^H NMR (400 MHz, DMSO-*d_6_*) δ: 3.05 (s, 6H, N(CH_3_)_2_), 6.78 (d, *J* = 8.3 Hz, 2H, Ar-H), 7.7 (d, *J* = 8.3 Hz, 2H, Ar-H), 7.77 (d, *J* = 12.6 Hz, 1H), 7.93 (d, *J* = 12.6 H, 1H), 8.13 (s, 2H, Benz-H), 13.66 (s, 1H, NH). IR (KBr, cm^−1^): 1349 (C-N), 1555 (C=N), 1641 (C=O), 3422 (N-H). MS *m*/*z* (% relative intensity, ion): 360 (35.53, M), 273 (100, ion). Elemental analysis for C_18_H_15_Cl_2_N_3_O. Calcd.: C, 60.02; H, 4.20; N, 11.66. Found: C, 60.19; H, 4.37; N, 11.52.

#### 4.1.2. Preparation of 4-(5-(5,6-Dichloro-1*H*-benzo[d]imidazol-2-yl)-4H-pyrazol-3-yl)-*N*,*N*-dimethylaniline **5**

A mixture of chalcone **4** (0.36 g, 1 mmol) and hydrazine hydrate (0.1 g, 2 mmol) in absolute ethanol (8 mL) was refluxed for 3 h, the reaction was monitored by TLC, and the reaction mixture was evaporated using rotavap to give the titled compound **5** as orange solids with mp = 216 °C (yield 70%).

Spectral data: ^1^H NMR (400 MHz, DMSO-*d_6_*) δ: 2.88 (s, 6H, N(CH_3_)_2_), 2.9 (s, 2H, pyrazole-H), 4.9 (s, 1H, NH), 6.72 (d, *J* = 8.2 Hz, 2H, Ar-H), 7.2 (d, *J* = 8.2 Hz, 2H, Ar-H), 7.6 (s, 1H, Benz-H), 7.88 (s, 1H, Benz-H). MS *m*/*z* (%): 496.26^+^ (18.05, M^+^), 497.9^+^ (6.75, M^+2^), 415^+^ (100). Elemental analysis for C_18_H_15_C_l2_N_5_. Calcd.: C, 58.08; H, 4.06; N, 18.81. Found: C, 58.16; H, 4.13; N, 18.73.

#### 4.1.3. Preparation of 2-Amino-6-(5,6-dichloro-1*H*-benzo[d]imidazol-2-yl)-4-(4-(dimethylamino)phenyl)nicotinonitrile **6**

A mixture of chalcone **4** (0.36 g, 1 mmol), malononitrile (0.2 g, 3 mmol), and ammonium acetate (0.62 g, 8 mmol) in absolute ethanol (10 mL) was heated for 3 h, after cooling, the reaction mixture was poured in ice water, neutralized with dilute HCl, filtered and washed with water to give the titled compound as black solids with mp = 168–170 °C (yield 70%).

Spectral data: ^1^H NMR (400 MHz, DMSO-*d_6_*) δ: 3.06 (s, 6H, N(CH_3_)_2_), 6.76 (s, 2H, NH_2_), 6.87 (d, *J* = 8 Hz, 2H, Ar-H), 7.61 (s, 1H, pyridine-H), 7.85 (d, *J* = 8 Hz, 2H, Ar-H), 8.06 (s, 2H, Benz-H). IR (KBr, cm^−1^): 1610 (C=N), 2214 (C≡N), 3225 and 3356 (NH_2_). MS *m*/*z* (%): 408.87^+^ (29.27, M^+^), 76.48^+^ (100). Elemental analysis for C_21_H_16_C_l2_N_6_. Calcd.: C, 59.59; H, 3.81; N, 19.85. Found: C, 59.52; H, 3.74; N, 20.02.

#### 4.1.4. Preparation of 2-Amino-6-(5,6-dichloro-1*H*-benzo[d]imidazol-2-yl)-4-(4-(dimethylamino)phenyl)-4*H*-pyran-3-carbonitrile **7**

A mixture of chalcone **4** (0.36 g, 1 mmol) and malononitrile (0.2 g, 3 mmol), and piperidine (3 drops) in absolute ethanol (10 mL) was refluxed for 24 h, the reaction mixture was evaporated using rotavap to give the titled compound **7** as black solids with mp = 150 °C (yield 70%).

Spectral data: ^1^H-NMR (400 MHz, DMSO-*d_6_*) δ: 2.9 (d, 1H, pyrane-H), 3.11 (s, 6H, N(CH_3_)_2_), 4.5 (s, 2H, NH_2_), 6.86 (d, *J* = 8.6 Hz, 2H, Ar-H), 7.2 (d, *J* = 8 Hz, 1H, pyrane-H), 7.84 (d, *J* = 8.6 Hz, 2H, Ar-H), 8.06 (s, 2H, Benz-H). ^13^C NMR (100 MHz, DMSO-*d_6_*) δ: 44.59 (CH_3_), 68.87 (CH), 112.20 (CH), 116.88 (CH_2_), 119.04 (CH), 133.78 (CH_2_), 133.78 (CH_2_), 154.82 (CH_2_), 159.32 (CH). IR (KBr, cm^−1^): 1611 (C=N), 2208 (C≡N), 3225 and 3424 (NH_2_). MS *m*/*z* (%): 372.4^+^ (16.88, M^+^), 374^+^ (6.15, M^+2^), 303.79^+^ (100). Elemental analysis for C_21_H_17_C_l2_N_5_O. Calcd.: C, 59.17; H, 4.02; N, 16.43. Found: C, 58.99; H, 3.92; N, 16.30.

#### 4.1.5. Preparation of 4-(3-(5,6-Dichloro-1*H*-benzo[d]imidazol-2-yl)-4,5-dihydroisoxazol-5-yl)-*N*,*N*-dimethylaniline **8**

A mixture of chalcone **4** (0.36 g, 1 mmol), hydroxylamine HCl salt (1 g, 33 mmol), and KOH (1.8 g, 33 mmol) in absolute ethanol (10 mL) was heated for 24 h, then the reaction mixture was poured in crushed ice water to give the titled compound (brown solids) with mp = 155–158 °C (yield 70%).

Spectral data: ^1^H NMR (400 MHz, DMSO-*d_6_*) δ: 2.81 (s, 6H, N(CH_3_)_2_), 3.0 (d, *J* = 6.5 Hz, 2H, isoxazoline-H), 5.75 (t, 1H, isoxazoline-H), 7.11 (d, *J* = 8.2 Hz, 2H, Ar-H), 7.27 (d, *J* = 8.2 Hz, 2H, Ar-H), 7.72 (s, 1H, Benz-H), 8.03 (s, 1H, Benz-H). MS *m*/*z* (%): 423.2^+^ (17.05, M^+^), 412.71^+^ (100). Elemental analysis for C_18_H_16_C_l2_N_4_O. Calcd.: C, 57.61; H, 4.30; N, 14.93. Found: C, 57.77; H, 4.11; N, 14.82.

#### 4.1.6. Preparation of 2-(2-Acetyl-5,6-dichloro-1*H*-benzo[d]imidazol-1-yl)-1-(4-bromophenyl)ethan-1-one **9**

Mixture of compound **3** (0.8 g, 3.5 mmol), 2-bromo-1-(4-bromophenyl)ethan-1-one (1.1 g, 4 mmol) and K_2_CO_3_ (0.83 g, 6 mmol) in acetone (10 mL) was stirred at room temperature for 3 h, after the reaction completion, the reaction mixture was filtered, washed by acetone, dried, then washed with water, filter and dried, to give the titled compound as gray solids with mp = 322 °C (yield 95%).

Spectral data: ^1^H NMR (400 MHz, DMSO-*d_6_*) δ: 2.68 (s, 3H, COCH_3_), 6.19 (s, 2H, COCH_2_), 7.63 (d, *J* = 8.2 Hz, 2H, Ar-H), 7.86 (s, 1H, Benz-H), 8.0 (s, 1H, Benz-H), 8.27 (d, *J* = 8.2 Hz, 2H, Ar-H). IR (KBr, cm^−1^): 1654, 1689 (C=O). MS *m*/*z* (%): 426.5^+^ (33, M^+^), 428^+^ (12.1^+^, M^+2^), 98.36^+^ (100). Elemental analysis for C_17_H_11_BrCl_2_N_2_O_2_. Calcd.: C, 47.92; H, 2.60; N, 6.57. Found: C, 47.82; H, 2.49; N, 6.41.

#### 4.1.7. Preparation of 3-(4-Bromophenyl)-7,8-dichloro-1-methylene-2-(p-tolyl)-1,2-dihydrobenzo[4,5]imidazo[1,2-a]pyrazine **10**

A mixture of compound **9** (0.426 g, 1 mmol) and *p*-toluidine (2.67 g, 25 mmol) in glacial acetic acid (8.5 mL) was refluxed for 24 h, after cooling, the reaction mixture was neutralized with K_2_CO_3_ solution, filtered and dried, to obtain the titled compound as brown solids with mp = 211 °C (yield 95%).

Spectral data: ^1^H NMR (400 MHz, DMSO-*d_6_*) δ: 2.02 (s, 3H, CH_3_), 7.09 (d, *J* = 8 Hz, 2H, Ar-H), 7.2 (d, *J* = 8 Hz, 2H, Ar-H), 7.4 (d, *J* = 8 Hz, 2H, Ar-H), 7.46 (d, *J* = 8 Hz, 2H, Ar-H), 8.95 (s, 2H, methelene-H), 8.69 (s, 1H, Benz-H), 8.86 (s, 1H, Benz-H), 9.7 (s, 1H, pyrazanobenzimidazole-H). ^13^C NMR (100 MHz, DMSO-*d_6_*) δ:20.9, 102.2, 112.4, 117.1, 119.5, 124.2, 128.9, 129.9, 130.5, 131.1, 131.5, 132.1, 134.2, 138.5, 145.1, 167.3. MS *m*/*z* (%): 373.86^+^ (34.2, M^+^), 375^+^ (11.7, M^+2^), 199.9 ^+^ (100). Elemental analysis for C_24_H_1__6_BrCl_2_N_3_. Calcd.: C, 57.89; H, 3.24; N, 8.45. Found: C, 57.72; H, 3.38; N, 8.59.

#### 4.1.8. Preparation of 3-(4-Bromophenyl)-7,8-dichloro-1-methylene-2-(o-tolyl)-1,2-dihydrobenzo[4,5]imidazo[1,2-a]pyrazine **11**

The reaction of compound **9** (0.426 g, 1 mmol) and *o*-toluidine (2.67 g, 25 mmol) in glacial acetic acid (8.5 mL) was performed under reflux for 24 h, the reaction mixture was poured in ice and neutralized with K_2_CO_3_ solution, then filtered and dried the formed precipitate to give the titled compound as brown solids with mp = 106 °C (yield 95%).

Spectral data: ^1^H NMR (400 MHz, DMSO-*d_6_*) δ: 2.06 (s, 3H, CH_3_), 7.18–7.41 (m, 4H, Ar-H), 7.58 (d, *J* = 7.4 Hz, 2H, Ar-H), 7.68 (d, *J* = 7.6 Hz, 2H, Ar-H), 8.13 (s, 1H, Benz-H), 8.25 (s, 1H, Benz-H), 8.85 (s, 2H, methelene-H), 8.91 (s, 1H, pyrazanobenzimidazole-H). MS *m*/*z* (%): 426.89^+^ (17.27, M^+^), 429.01^+^ (6.15, M^+2^), 105.77^+^ (100). Elemental analysis for C_24_H_16_BrCl_2_N_3_. Calcd.: C, 57.89; H, 3.24; N, 8.45. Found: C, 57.81; H, 3.19; N, 8.58.

#### 4.1.9. Preparation of 2-Benzyl-3-(4-bromophenyl)-7,8-dichloro-1-methylene-1,2-dihydrobenzo[4,5]imidazo[1,2-a]pyrazine **12**

Refluxing of a mixture of compound **9** (0.426 g, 1 mmol) and benzylamine (2.67 g, 25 mmol) in glacial acetic acid (8.5 mL) was performed for 24 h, the reaction mixture was poured in ice, neutralized with K_2_CO_3_ solution, then the formed precipitate was filtered and dried, to give the desired compound as dark green precipitate with mp = 85 °C (yield 50%).

Spectral data: ^1^H NMR (400 MHz, DMSO-*d_6_*) δ: 4.64 (s, 2H, CH_2_-Ph), 7.26–7.48 (m, 5H, Ar-H), 7.67 (d, *J* = 7.2 Hz, 2H, Ar-H), 7.8 (d, *J* = 7.2 Hz, 2H, Ar-H), 8.07 (s, 2H, Benz-H), 8.52 (s, 2H, methelene-H), 8.77 (s, 1H, byrazanobenzimidazole-H). ^13^C NMR (100 MHz, DMSO-*d_6_*) δ: 64.73 (CH_2_), 101.23 (CH), 112.02 (CH), 115.08 (CH), 120.65 (CH), 123.35 (CH_2_), 124.97 (CH_2_), 127.31 (CH), 128.93 (CH), 131.08 (CH), 132.34 (CH), 136.66 (CH_2_), 139.54 (CH_2_), 140.25 (CH), 142.96 (CH_2_), 144.21 (CH_2_), 162.19 (CH). MS *m*/*z* (%): 407.11^+^ (52.18^+^, M^+^), 407.27^+^ (28.7, M^+2^), 104.31^+^ (100). Elemental analysis for C_24_H_16_BrCl_2_N_3_. Calcd.: C, 57.89; H, 3.24; N, 8.45. Found: C, 57.81; H, 3.39; N, 8.28.

#### 4.1.10. Preparation of 3-(4-Bromophenyl)-7,8-dichloro-1-methylbenzo[4,5]imidazo[1,2-a]pyrazine **13**

A mixture of compound **9** (0.426 g, 1 mmol) and ammonium acetate (2 g, 25 mmol) in glacial acetic acid (7.5 mL) was heated for 24 h, after the reaction performance, the resulted solution was poured into ice, neutralized with ammonia then the formed solid was filtered, washed with water, and dried, to give the titled compound as puff precipitate with mp = 290 °C (yield 60%).

Spectral data: ^1^H-NMR (400 MHz, DMSO-*d_6_*) δ: 1.25 (s, 3H, CH_3_), 7.77 (d, *J* = 7.7 Hz, 2H, Ar-H), 8.1 (d, *J* = 7.7 Hz, 2H, Ar-H), 8.33 (s, 1H, Benz-H), 8.93(s, 1H, Benz-H), 9.68 (s, 1H, pyrazanobenzimidazole-H). MS *m*/*z* (%): 497.7^+^ (16.59, M^+^), 499.6 (5.3, M^+2^) 159.17^+^ (100). Elemental analysis for C_17_H_10_BrCl_2_N_3_. Calcd.: C, 50.16; H, 2.48; N, 10.32. Found: C, 50.25; H, 2.58; N, 10.11.

#### 4.1.11. Preparation of 2-(4-Bromophenyl)-7,8-dichlorobenzo[4,5]imidazo[1,2-a]pyridin-4-ol **14**

A mixture of compound **9** (0.426 g, 1 mmol) and K_2_CO_3_ (0.7 g, 5 mmol) in *n*-butanol (10 mL) was refluxed for 6 h, the reaction mixture was filtered, washed with butanol, dried, then washed with glacial acetic acid solution, filtered, and dried to give the titled compound as puff precipitate with mp = >300 °C (yield 50%).

Spectral data: ^1^H-NMR (400 MHz, DMSO-*d_6_*) δ: 7.19 (d, *J* = 7.7 Hz, 2H, Ar-H), 7.3 (d, *J* = 7.7 Hz, 2H, Ar-H), 7.74 (s, 2H, Benz-H), 8.11 (s, 1H, pyridinonobenzimidazole-H), 8.85 (s, 1H, pyridinobenzimidazole-H), 9.07 (s, 1H, OH). IR (KBr, cm^−1^): 1219–1296 (C-O), 1557–1642 (C=C, C=N), 3448 (OH). MS *m*/*z* (%): 497.8^+^ (5.55, M^+^), 51.31^+^ (100). Elemental analysis for C_17_H_9_BrC_l2_N_2_O. Calcd.: C, 50.04; H, 2.22; N, 6.86. Found: C, 50.14; H, 2.01; N, 6.75.

### 4.2. Cytotoxic Assay

Cell line: Mammary gland breast cancer (MCF-7). The cell line was obtained from ATCC via Holding company for biological products and vaccines (VACSERA), Cairo, Egypt. Doxorubicin was used as a standard anticancer drug for comparison. Chemical reagents: the reagents RPMI-1640 medium, MTT and DMSO (Sigma, St. Louis, MO, USA), Fetal Bovine serum (GIBCO, Oxford, UK) [47]. The cell line elucidated above were used to determine the inhibitory effects of compounds on cell growth using the MTT assay. Cell lines were cultured in RPMI-1640 medium with 10% fetal bovine serum, then added the antibiotics were 100 units/mL penicillin and 100 µg/mL streptomycin at 37 °C in a 5% CO_2_ incubator. The cell lines were seeded in a 96-well plate at a density of 1.0 × 104 cells/well at 37 °C for 48 h under 5% CO_2_. After incubation, the cells were treated with different concentrations of compounds and incubated for an additional 24 h. After 24 h of drug treatment, 20 µL of MTT solution at 5 mg/mL was added and incubated for 4 h. A volume of 100 µL from dimethyl sulfoxide (DMSO) is added to each well to dissolve the purple formazan formed, then the absorbance was measured at wavelength 570 nm using a plate reader (EXL 800, Jersey City, NJ, USA). The relative cell viability in percentage was calculated as (A570 of treated samples/A570 of the untreated sample) × 100 [48].

### 4.3. Pin1 Inhibition Assay

The ability of the synthesized compounds to inhibit the Pin1 activity was evaluated using the SensoLyte^®^ Green Pin1 Assay Kit. This assay is based on using a fluorogenic substrate which was pretreated to convert it into the cis isomer. Pin1 can change this substrate into the trans isomer that is easily cleaved to generate a fluorescent signal. Fluorescence is then monitored at υ(Ex/Em) = 490/520 nm and hence the increase in fluorescence intensity is directly proportional to the Pin1 activity. The bioassay was performed by following steps:Step 1: Prepare working solutions composed of Pin1 substrate solution (Dilute Pin1 substrate 1:100 with the assay buffer in which the working substrate solution of 1 µM. An amount of 50 µL of this diluted substrate is enough for one-well reaction), Recombinant Pin1 diluent (Dilute human enzyme with 190 μL of the assay buffer to a final concentration of 50 μg/mL), Pin1 Inhibitor (Dilute Pin1 Inhibitor 1:50 with the assay buffer to a final concentration of 1 mM), Reaction Developer (Dilute Pin1 Developer 1:100 with the assay buffer and each assay well requires 30 µL of the prepared enzyme developer).Step 2: Set up the enzymatic reaction through the following procedure: add 10 µL of test compounds and 10 µL enzyme solution to the microplate wells, then establishing the following control wells at the same time: Positive control contains Pin1 enzyme without test compound;Inhibitor control contains Pin1 and Tannic acid;Vehicle control contains enzymes and vehicles used in delivering test compounds like DMSO;Test compound control contains assay buffer and test compound;Background developer control contains assay buffer without enzyme.Then using assay buffer, bring the total volume of all controls to 20 µL and Add Pin1 Developer solution (from 1.4) at 30 μL/well.Step 3: Run the enzymatic reaction by adding 50 µL of Pin1 substrate (from 1.1) solution into each well and mixing reagents completely by shaking the plate gently for no more than 30 sec, then measuring the fluorescence signal.Step 4: Data Analysis: Substrate will change its conformation into the more stable trans form slower rate, so readings from wells containing background developer control must be subtracted from other controls and test compound reading and all fluorescence readings are expressed in relative fluorescence units (RFU).

### 4.4. Experimental NMR for Drug-Design

The expression and purification of ^15^N-labeled Pin1 was performed according to the known protocol [49,50]. The enzyme sample reached a final concentration of 200 µM in PBS buffer at pH = 6.5. Compound number **12** was dissolved in DMSO and added to the protein to achieve a final concentration of 1 mM in the NMR tube. The ^15^N-^1^H-HSQC experiments have been performed for the protein alone or with the compound on a Varian-900MHz-cold probe at the University of Colorado, Denver, USA. The number of points used in the indirect dimension were 128 with 32 scans. The spectra were processed using NMRPipe [51] and figures were created using Sparky [52] software.

### 4.5. Apoptosis Assay

Apoptosis detection was performed by using Annexin V-FITC and PI apoptosis kit (eBioscience^TM^, San Diego, CA, USA). Mcf cells were plated at a 600,000 cells/mL density onto a six well plate. After 24 h of incubation, the cells were treated with compound **8** at 10 mg/mL. Cells grown in media containing an equivalent amount of DMSO served as solvent control. After 24 h, the cells were stained with an Annexin V-FITC conjugate and propidium iodide (PI), and the percentage of apoptotic, necrotic, and living cells was determined according to the protocol provided by the Annexin V-FITC and PI apoptosis kit. The cells’ emitted fluorescence was analyzed by flow cytometry (NovoCyte, ACEA Biosciences Inc., San Diego, CA, USA) through the NovoExpress 1.3.0 software (ACEA Biosciences Inc., San Diego, CA, USA), acquiring 1 × 104 events per sample using the population plot “dot plot”, where each point corresponds to a single event with a specific fluorescence signal about the axes; Annexin V-FITC green fluorescence in abscissa vs. PE red fluorescence inordinate.

#### Measurement of the Effect of Compound **8** on the Expression of Apoptotic and Anti-apoptotic Markers in MCF-7 Cell Line

The level of BAX, Bcl-2, cytochrome C, and caspases 6, 7, and 9 were estimated using ELISA colorimetric kits; for BAX (DRG^®^ Human Bax ELISA, EIA-4487), Bcl-2 (Zymed^®^ Bcl-2 ELISA Kit), cytochrome c (Abcam cytochrome c Human ELISA kit, ab119521), caspase-6 (Human CASP6, ELISA Kit, MBS2533323), caspase-7 (Caspase-7 (Human) ELISA Kit, E4295-100), caspase-9 (DRG^®^ Caspase-9 (human) ELISA, EIA-4860). The process of the used kits was performed according to the manufacturer. Briefly, MCF-7 were cultured in RPMI1640 including 10% fetal bovine serum at 37 °C then treated with compound **8** and lysed with cell extraction buffer. This lysate was diluted in Standard Diluent Buffer all over the assay, then by using a microplate reader set at 450 nm, the optical density of each well was determined within 30 min to determine the human active BAX, Bcl-2, cytochrome C, caspases 6, 7 and 9 content.

### 4.6. Intracellular ROS Accumulation Assay

The Human ROS ELISA Kit is based on standard sandwich enzyme-linked immunosorbent assay technology. In which, the anti-Human ROS specific antibody has been pre-coated onto a 96-well plate and the human ROS present in the standards/samples bind to the capture antibody. Then biotinylated anti-Human ROS detection antibody is added to form an Ab-Ag-Ab sandwich. After a washing step, streptavidin-HRP is added and the unbound conjugate is removed with wash buffer. Subsequently, the addition of HRP substrate, TMB, results in the production of a blue-colored product that changes to yellow after the addition of acidic Stop Solution, and the density of yellow color is directly proportional to the amount of Human ROS captured on the plate. The process performed in this assay was summarized as following: washing the plate twice before adding standard, sample and control (zero) wells, adding 100 μL standard or sample to each well for 90 min at 37 °C and adding 100 μL Biotin-detection antibody working solution to each well for 60 min at 37 °C, then aspirate and wash 3 times and add 100 μL SABC working solution to each well. Then Incubate for 30 min at 37 °C, aspirate and wash 5 times. After that adding 90 μL TMB substrate, incubating 15–30 min at 37 °C, adding 50 μL Stop Solution and reading at 450 nm immediately. Finally, calculation of the results.

### 4.7. Computatinal Studies

MOE (Chemical Computing Group Inc. software 2014, Montreal, QC, Canada) was used to build the 3D structures of some selected substituted benzimidazole **5**, **8**, and **12** in their neutral forms. The selected compounds represent the best anti-breast cancer activity.

The lowest energy conformers of new analogues ‘global-minima’ were docked into the binding pocket of Pin1 (PDB code: 4TYO) [32] which was obtained from the Protein Data Bank of Brookhaven National Laboratory. The hydrogens were added, then enzyme structure was subjected to a refinement protocol where the constraints on the enzyme were gradually removed and minimized until the RMSD gradient was 0.01 kcal/mol Å. Energy minimization was carried out using the molecular mechanics force field ‘AMBER’. For each benzimidazole derivative, energy minimizations (EM) were performed using 1000 steps of steepest descent, followed by conjugate gradient minimization to a RMSD energy gradient of 0.01 kcal/mol Å. The active site of the enzyme was detected using a radius of 10.0 Å around MTX. Energy of binding was calculated as the difference between the energy of the complex and individual energies of the enzyme and ligand [53,54]. The compounds under study underwent flexible alignment experiment using ‘Molecular Operating Environment’ software (MOE of Chemical Computing Group Inc., on a Core i7 2.3 GHz workstation). The molecules were constructed using the Builder module of MOE. Their geometry was optimized by using the MMFF94 forcefield followed by a flexible alignment using systematic conformational search and the Lowest energy aligned conformers were identified. ADMET calculations (Absorption, Distribution, Metabolism, Excretion, and Toxicity) were determined using implemented tool in MOE, 2014. The molecular structure of the tested flavonoids was constructed from fragment libraries MOE [55,56,57]. For each analog, the partial atomic charges were assigned using the semi-empirical mechanical calculation “AM1” method implemented in the program [58]. The conformational search was done. All the conformers were minimized until the RMSD deviation was 0.01 kcal/mol, then subjected to surface mapping, color-coded: pink, hydrogen bond; blue, mild polar; green, hydrophobic.

## Data Availability

Not available.

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
