# Peer review of "New Benzimidazoles Targeting Breast Cancer: Synthesis, Pin1 Inhibition, 2D NMR Binding, and Computational Studies"

_molecules, 2022, doi:10.3390/molecules27165245_

Round 1

Reviewer 1 Report

To Author:

The topic of the article “New Benzimidazoles Targeting Breast Cancer: Synthesis, Pin1  Inhibition, 2D NMR Binding, and Computational Studies” is interesting. The search for new active substances that could be used to treat cancer is a current and needed issue. Especially at a time when there is a continuous increase in the incidence of various types of cancer. The authors present proper strategy from design, synthesis of new benzimidazole derivatives, characterization of their chemical properties of them, and finally the biological evaluation. The tests on cell lines make an invaluable positive work contribution. In my opinion, the quality of the presentation the results of the research and have to be improved. I would reconsider the publishing this article in Molecules but only after major revision according to the comments listed below:

Comments to Main text:

1.      You tested the cytotoxic activity of compounds against MCF-7 cell line and then you selected the compounds to evaluate the cytotoxicity on a normal Human lung fibroblast (WI38) cell line. Why you did not check the cytotoxicity on WI38 for compounds 7 and 4?

2.      In Table 1 with physicochemical properties of synthesized compounds please add the molecular weight of them. It will be easier to read and analyze the MS spectra.

3.      Figure 1 should be revised in better resolution and the windows watermark should be removed.

4.      The structure of compound 8 on 9 page at line 236 is not necessary. It will be better to remove it because it brings nothing to work. In addition, it is a duplication of information from the previous part of the article.

5.      The size font of the legend in Figure 2 must be larger and the date should be removed. You can move the legend to the empty space on the graph.

6.      The date and name of the sample in Figure 4 should be removed.

7.      All graphs in Figure 5 should be presented in the same way, font size, and type, proportions, etc. Also the table should be include as a separate table.

8.      In chapter 2.10.2. Molecular docking study, the name and location of amino acid in protein sequence should be written in the same way in all text. I recommend using eg.: ‘Phe134’. Please verify all records for the same.

9.      The mean ± SD values should be given with the same accuracy, in table in figure 5.

10.  Please revise the text and correct any editorial errors such as: missing ‘space’, too many “[”, missing dot, not necessary hyphen or minus, repeating the same word (e.g. 'part' at line 110),

11.  Please do not use informal signs like ‘&’, instead use ‘and’. There are a lot of abbreviation, phrases or words that are undesirable in the scientific text and must be changed. At line:

-          115, 139 - ‘amm acetate’ I think it should be ‘ammonium acetate’;

-          123 – ‘gp’ I think it should be ‘group’;

-          127 – ‘douplet’ should be ‘doublet’;

-          137 – the ion charge should be written in upper index…. and so far and so on.

So my advice is: read carefully all manuscript one more time and revise all such bugs. 

Comments to Supplementary:

1. Figures with spectra should be devoid of detailed descriptions coming directly from the devices, eg the path of the file, etc. The Figures must be presented with a better resolution. In some of them, the numerous data are hard to read. It looks like somebody made the ‘print screen' and paste it directly to the file, it is not professional.

2. The 1H 13and C shifts on NMR spectra have to be presented in the table or mentioned in the caption of the figure with a signed atom number.

3. At the MS spectra each peak must possess the charge.

4.The IR spectra require deeper analysis and assigning the signals to the appropriate vibrations in the molecule.

5.The Supplementary file needs to have a table of contents. All Figures and Tables should be numbered with proper caption.

Author Response

To Author:

The topic of the article “New Benzimidazoles Targeting Breast Cancer: Synthesis, Pin1 Inhibition, 2D NMR Binding, and Computational Studies” is interesting. The search for new active substances that could be used to treat cancer is a current and needed issue. Especially at a time when there is a continuous increase in the incidence of various types of cancer. The authors present proper strategy from design, synthesis of new benzimidazole derivatives, characterization of their chemical properties of them, and finally the biological evaluation. The tests on cell lines make an invaluable positive work contribution. In my opinion, the quality of the presentation the results of the research and have to be improved. I would reconsider the publishing this article in Molecules but only after major revision according to the comments listed below:

Thanks a lot for the positive comments. Below we reply to all the comments one by one. All modifications in the manuscript are highlighted in yellow color and tracking is kept activated.

Comments to Main text:

  1. You tested the cytotoxic activity of compounds against MCF-7 cell line and then you selected the compounds to evaluate the cytotoxicity on a normal Human lung fibroblast (WI38) cell line. Why you did not check the cytotoxicity on WI38 for compounds 7 and 4?

Thanks a lot for the comment.

We selected the most active compounds which display higher activity against breast cancer cell lines (5, 8, and 12) for further studies to evaluate their cytotoxicity and to test their safety. While compounds 4 and 7 are less active candidates.

  1. In Table 1 with physicochemical properties of synthesized compounds please add the molecular weight of them. It will be easier to read and analyze the MS spectra.

A column of molecular weight of the compounds has been added.

  1. Figure 1 should be revised in better resolution and the windows watermark should be removed. 

We revised the figure and improved the resolution using SPARKY-NMR software and we also removed the watermark.

  1. The structure of compound 8 on the 9 page at line 236 is not necessary. It will be better to remove it because it brings nothing to work. In addition, it is a duplication of information from the previous part of the article.

The structure has been removed as suggested.

  1. The size font of the legend in Figure 2 must be larger and the date should be removed. You can move the legend to the empty space on the graph. 

We adjusted that.

  1. The date and name of the sample in Figure 4 should be removed.

The date and the name of the sample in Figure 4 have been removed.

  1. All graphs in Figure 5 should be presented in the same way, font size, and type, proportions, etc. Also the table should be include as a separate table.

We created a new figure with exact same fonts and types. We also did the same for figure 6 to match this one.

  1. In chapter 2.10.2. Molecular docking study, the name and location of amino acid in protein sequence should be written in the same way in all text. I recommend using eg.: ‘Phe134’. Please verify all records for the same.

Lys 63 and Arg 69 in different positions of manuscript are modified in to Lys63 and Arg69

  1. The mean ± SD values should be given with the same accuracy, in table in figure 5.

We fixed that

  1. Please revise the text and correct any editorial errors such as: missing ‘space’, too many “[”, missing dot, not necessary hyphen or minus, repeating the same word (e.g. 'part' at line 110),

Many writing errors are corrected along the text and the word (part) is removed. We also did a lot of minor grammatical modifications.

  1. Please do not use informal signs like ‘&’, instead use ‘and’. There are a lot of abbreviation, phrases or words that are undesirable in the scientific text and must be changed. At line:

-          115, 139 - ‘amm acetate’ I think it should be ‘ammonium acetate’;

-          123 – ‘gp’ I think it should be ‘group’;

-          127 – ‘douplet’ should be ‘doublet’;

-          137 – the ion charge should be written in upper index…. and so far and so on. 

So my advice is: read carefully all manuscript one more time and revise all such bugs. 

The informal sign ‘&’ was replaced with ‘and’.

The abbreviations ‘amm acetate, gp and douplet ’  got changed to ‘ammonium acetate, group and doublet’, respectively.

The ion charge is written in upper index

We fixed many typos and grammar mistakes all over the manuscript. We also asked a native speaker to go through the whole manuscript

 Comments to Supplementary:

  1. Figures with spectra should be devoid of detailed descriptions coming directly from the devices, eg the path of the file, etc. The Figures must be presented with a better resolution. In some of them, the numerous data are hard to read. It looks like somebody made the ‘print screen' and paste it directly to the file, it is not professional.

The detailed description of figures with spectra is removed, and the resolution has been improved.

  1. The 1H and 13C shifts on NMR spectra have to be presented in the table or mentioned in the caption of the figure with a signed atom number.

 A list of spectroscopic spectra of the compounds is added

  1. At the MS spectra each peak must possess the charge. 

The correct positive charge of each peak has been added.

  1. The IR spectra require deeper analysis and assigning the signals to the appropriate vibrations in the molecule.

This has been fixed.

5. The Supplementary file needs to have a table of contents. All Figures and Tables should be numbered with proper caption.

A list of manuscript figures and tables has been added

Reviewer 2 Report

In the paper “New Benzimidazoles Targeting Breast Cancer: Synthesis, Pin1 Inhibition, 2D NMR Binding and Computational Studies” authors have synthesized a series of benzimidazole derivatives conjugated with six and five membered heterocyclic rings. I found the manuscript interesting but not written properly. This manuscript required a major revision before considering for publication.

1.      Line 15, ESI-MS is not a spectroscopic technique. Revise the sentence.

2.      Re-write abstract in a specific order. Background, objective of this study, methods, results and conclusion.

3.      2D NMR 15N-1H HSQC NMR, 2D NMR is unnecessary written here.

4.      Figures are not properly arranged. Figures 13, 14, and 15 should be in supplementary information.

5.      Arrange figures properly  and remove chart 1 and etc. Group sub figures in a figure.

6.      Write Figure caption of each figure in the supplementary information or a tabular format on first page of supplementary information indicating the caption of each figure with page number.

7.      There is no flow at all from introduction to conclusion in the manuscript. Arrange figures properly.

8.      Why HSQC spectrum in blue looks unfolded protein structure in the solution. Red one looks properly folded.

Author Response

In the paper “New Benzimidazoles Targeting Breast Cancer: Synthesis, Pin1 Inhibition, 2D NMR Binding and Computational Studies” authors have synthesized a series of benzimidazole derivatives conjugated with six and five membered heterocyclic rings. I found the manuscript interesting but not written properly. This manuscript required a major revision before considering for publication. 

Thanks a lot for the positive comments. Below we reply to all the comments one by one. All modifications in the manuscript are highlighted in yellow color and tracking is kept activated.

  1. Line 15, ESI-MS is not a spectroscopic technique. Revise the sentence.

The sentence has been corrected

  1. Re-write abstract in a specific order. Background, objective of this study, methods, results and conclusion.

The abstract has been modified and we fixed writing issues and added the conclusion part.

  1. 2D NMR 15N-1H HSQC NMR, 2D NMR is unnecessary written here.

2D NMR was removed

  1. Figures are not properly arranged. Figures 13, 14, and 15 should be in supplementary information.

Figures 13, 14, and 15 are removed

  1. Arrange figures properly and remove chart 1 and etc. Group sub figures in a figure.

Chart 1 has been moved to the supplementary (named chart s1). Figures were arranged based on modifications. We also modified extensively figures 5 and 6.

  1. Write Figure caption of each figure in the supplementary information or a tabular format on first page of supplementary information indicating the caption of each figure with page number.

A table of figure captions and page numbers was added to the first page of supplementary information.

  1. There is no flow at all from introduction to conclusion in the manuscript. Arrange figures properly.

The manuscript was revised carefully to assure the flow of ideas from the introduction to conclusion. Moreover, the figures were arranged as requested.

  1. Why HSQC spectrum in blue looks unfolded protein structure in the solution. The red one looks properly folded.

The HSQC in blue is showing dispersed peaks which indicates that the protein is still folded. We only see some extra peaks which can be from a slow exchange between free and bound Pin1.

Round 2

Reviewer 1 Report

Dear Authors,

The manuscript: 'New Benzimidazoles Targeting Breast Cancer: Synthesis, Pin1 Inhibition, 2D NMR Binding and Computational Studies' in current quality is ready to publish. However, I cannot understand why the authors decided to publish this paper to Magnetochemistry Journal. This work for me is bio-chem-medical. The research is described with chemical methods, including synthesis, and research on cell lines. I believe that the article should be published in a journal on a suitable subject, for example by MDPI other Journals like: BioChem, Biomolecules, Current Issues in Molecular Biology.

Reviewer 2 Report

Authors have provided appropriate responses to my comments and significantly improved the manuscript. I recommend this manuscript to be published in this Journal.